# Were the socio-economic determinants of municipalities relevant to the increment of COVID-19 related deaths in Brazil in 2020?

**Julio Castro-Alves●\*, Lídia Santos Silva●, João Paulo Lima, Marcelo Ribeiro-Alves●**

Instituto Nacional de Infectologia Evandro Chagas, Fiocruz, Rio de Janeiro, RJ, Brazil

\* Julio.castro.alves.lima@gmail.com

## Abstract

### Background

The COVID-19 pandemic in Brazil has been showing a pattern of distribution of related deaths associated with individual socioeconomic status (SES). However, little is known about the role of SES in the distribution of the mortality rate in different population, from an ecological perspective.

### Objective

The objective of this study was to evaluate the role of socioeconomic factors in the distribution of the COVID-19-related mortality rate among Brazilian municipalities in 2020.

### Methods

We conducted a retrospective, cross-sectional, observational, population-wide, and ecological study, using data of COVID-19-related deaths from the Influenza Epidemiological Surveillance Information System database (SIVEP-Gripe) and SES from the Social Vulnerability Index (SVI), the Human Development Index (HDI), the Geographic Index of the Socio-economic Context and Social Studies (GeoSES), and 2010 Demographic Census (IBGE/Brazil). We computed crude, age- and sex-standardized, and the latter offset by the time of exposure to the epidemic mortality rates. To determine socioeconomic factors associated with mortality rates we used log-linear models with state codes as a random effect and Haversine variance-covariance matrix.

### Results

191,528 deaths were related to COVID-19 and distributed in 4,928 (88.55%) Brazilian municipalities. Whatever the socioeconomic indexes used, the R2 were very small to explain SMRT. Consistent across all socioeconomic indexes used, high-income, more educated, and well infrastructure municipalities generally had higher mortality rates.

**Data Availability Statement:** The data underlying the results presented in the study are available from Influenza Epidemiological Surveillance Information System database, InfoGripe, Brazilian

Ministry of Health (https://opendatasus.saude.gov.br/dataset/srag-2020).

**Funding:** This research was funded by Inova-Fiocruz Program (VPPCB-005-FIO-20-2-45, https://portal.fiocruz.br/programa-inova-fiocruz) to M. Ribeiro-Alves and Inova-Fiocruz Program (VPPCB-005-FIO-20-2-61, https://portal.fiocruz.br/programa-inova-fiocruz) to J. Castro-Alves. The funders had no role in the conceptualization, analysis, interpretation, or decision to publish this manuscript.

**Competing interests:** The authors have declared that no competing interests exist.

## Conclusion

Excluding the effect of demographic structure and pandemic timing from mortality rates, the contribution of SES to explain differences in COVID-19-related mortality rates among municipalities in Brazil became very low. The impact of SES on COVID-19-related mortality may vary across levels of aggregation. Urban infrastructure, which includes mobility structures, more complex economic activities and connections, may have influenced the average municipal death rate.

## Introduction

In December 2019, in Wuhan (China), the first cases of infection with the SARS-CoV-2 virus emerged [1]. By January 30th, 2020, the date it was declared by the World Health Organization (WHO) as a Public Health Emergency of International Importance [2], the new coronavirus had already spread to 18 countries. The infection and deaths from COVID-19 accelerated, forcing the WHO, on March 11th, 2020, to declare a situation of Pandemic [2].

Although poverty and inequality have not been prominent causal factors in epidemiological transmission for COVID-19, the social science literature has pointed to many ways the two former can be expected to increase vulnerability to coronavirus infection [3]. From the perspective of social epidemiology [4], a system of inequality containing social aspects such as education, material conditions and infrastructure, poor working conditions, racial segregation, and gender inequality acts concomitantly in cascade influencing the disease's biology and deaths. A general premise of this line of thinking is that social inequalities would shape individuals' exposure and susceptibility to SARS-CoV-2 infection and COVID-19-related deaths, generating health inequalities. Based on this thinking, Clouston *et al.* (2021) proposed a narrative for the spread pattern in the COVID-19 epidemic. In the early epidemic period, the SARS-CoV-2 virus emerged and diffused through the population [5]. At first, elites and high socioeconomic status (SES) communities would become infected because they have more opportunities to travel abroad and, consequently, bring the virus to their countries. In addition, the limited knowledge accumulated about the new disease engenders ineffective protocols for dealing with it. Thus, leading initial cases to occur in higher SES locations and, gradually, spreading to the community, including lower SES. After that, as a public health response arises, communities begin to implement social distancing strategies to control the risk of viral spread and infection, inequalities emerge and grow. Due to the distinct ability to mobilize resources in containing the spread of the virus, high- and low-SES communities would face divergent epidemic situations. There would be a reduction in prevalence and a gradual decrease in the risk of COVID-19-related death for the former. For the latter, the burden of disease would increase, and they would experience more and more cases of COVID-19. The health inequalities would arise as an unequal diffusion of health innovations. As the pandemic progresses, exposure to SARS-CoV-2 virus increases in neighborhoods with overcrowded households and poor access to drinking water; or due to differences in job quality, such as informal jobs with low wages, few benefits, limited hours of work, and no chance to work remotely; and susceptibility increases due to chronic comorbidities that follow a pattern motivated by social disparities. So, the presence of a health gradient against racial minorities, black/brown people [6–9] and vulnerable populations [8–11] become even more evident. Finally, the reduction of health inequalities would occur due to the increasing

access to health knowledge among populations, just as with the development of effective vaccines and the eventual disease elimination.

The Clouston *et al.* (2021) narrative about the rise and fall of health inequalities during the COVID-19 epidemic and their association with SES inequalities is quite consistent considering the Brazilian experience regarding individual data [5, 12–17]. However, about the municipalities, the inter-municipal or ecological dynamics of spread, another narrative seems to have prevailed. Being a continental and very unequal country, we had in Brazil not one, but several patterns of COVID-19 spread., The pandemic migrated from the wealthier southeastern region to the poorer northern region. The SARS-CoV-2 virus circulated first among densely populated municipalities (capital cities) and, gradually, due to their dense urban network, that also connects to other municipalities, spread to other capital cities and, from these, to the interior of the country [18–20]. Consequently, income and urban infrastructure, including mobility structures, roads, transportation, more complex economic activities and connections, have influenced the average municipal death rate. In other countries, such as France, a similar inter-municipal pattern has been observed [10]. The role of SES as a determinant of mortality inequality during the COVID-19 pandemic in Brazil seems to diverge from the norm depending on the level of the analysis, which requires an understanding of this relationship from an ecological perspective. This study aimed to examine the role of socioeconomic factors in the distribution of the COVID-19-related mortality rate among Brazilian municipalities in 2020. To this end, we calculate age- and sex-standardized, time-adjusted mortality rates of COVID-19 for all municipalities from February 20th to December 31st, 2020. Furthermore, we develop a robust analysis comparing different definitions of mortality rate and a broad set of available SES indices.

This work brings some novel and robust information to the existing literature, mainly associated with the methodology analysis. So far, we are the first study to our knowledge to address the impact of different mortality definitions on the association of COVID-19 deaths with SES. Most ecological studies used crude mortality rates without examining the impact of this choice on results. In addition, there is a lack of studies investigating the role of socioeconomic inequalities using a robust set of SES indices to explain differences in mortality rates among regions. We examined all available SES data in Brazil. We used a nationwide populational dataset, whereas most literature used samples from municipalities, states, or compositions. The latter can be seen in Brazilian studies, but also in other countries. Finally, considering statistical modeling aspects, our models included a broad set of variables as confounders to test the robustness of SES. It included the timing of the epidemic in each municipality, the spatial structure of correlation between municipalities, and, as independent variables, natural causes of virus spread, and health services coverage.

## Material and methods

### Data

We conducted a retrospective, cross-sectional, observational, population-wide, and ecological study of the relationship between socioeconomic status and the COVID-19-related mortality rate by municipality of residence. Below, we briefly describe the data and information sources used.

**Health data.**   The information on the number of COVID-19-related deaths and the pandemic time (date of first symptoms for the first recorded case) by the municipality of residence was extracted from the Influenza Epidemiological Surveillance Information System database (SIVEP-Gripe), as of January 11, 2021. The data underlying the results presented in the study are available from Influenza Epidemiological Surveillance Information System database,

InfoGripe, Brazilian Ministry of Health (https://opendatasus.saude.gov.br/dataset/srag-2020).
SIVEP-Gripe is an official open-source national surveillance database used to monitor severe
acute respiratory infections in Brazil and resulting hospital admissions, established and
updated by the Brazilian Ministry of Health (MoH, Brazil). From this, we selected only patients
with a confirmed diagnosis of COVID-19 (laboratory, clinical, clinical-epidemiological, or
imaging criteria) with clinical evolution until COVID-19-related death from February 20 (first
confirmed case of COVID-19 in Brazilian territory) to December 31, 2020. Next, the number
of deaths and the pandemic time until December 31, 2020, were grouped by the municipality
of residence of each registrant.

**Socioeconomic data.**   For us to assess the robustness of our observed results, we used four
different ways to quantify the socioeconomic status. These were: the Social Vulnerability Index
(SVI) (http://ivs.ipea.gov.br/index.php/pt/), developed by the Institute for Applied Economic
Research (IPEA, Brazil) (https://www.ipea.gov.br/portal/index.php?option=com_content&id=
19153); the Human Development Index (HDI), developed by the United Nations (UN); the
Geographic Index of the Socioeconomic Context and Social Studies (GeoSES) [21] (https://
doi.org/10.1371/journal.pone.0232074.s003), developed by the Support Program for Institu-
tional Development of the Unified Health System (PROADI-SUS) and coordinated by the
Unified Health System (SUS) Monitoring and Evaluation Department (DEMAS/MoH, Brazil);
and, finally, a selection of socioeconomic indices/rates among many developed by the Brazilian
Institute of Geography and Statistics (IBGE, Brazil). The four socioeconomic approaches were
created from variables from the 2010 Demographic Census covering all Brazilian municipali-
ties (IBGE, Brazil). Below, we will further detail each of these theoretical ways of conceiving
the socioeconomic status of Brazilian municipalities. Detailed information about each index
composition can be found at S2 Table.

The SVI is based on the recognition that social vulnerabilities arise from broader social pro-
cesses, which individuals have little autonomy to change, and therefore, it would be up to the
State to change these conditions through public policies. In this sense, the index is separated
into three large sets of assets, whose possession, or deprivation, would determine the welfare
conditions of the populations: Urban Infrastructure, Human Capital, and Income and Labor.
Each of these dimensions varies from 0 to 1, where 0 corresponds to the ideal situation and 1
corresponds to the worst situation, that is, municipalities with the SVI close to 1 are in the situ-
ation of maximum social vulnerability. The aggregated SVI index would be a composition of
these three dimensions, varying in the same range.

The Human Development Index (HDI) is intended to measure the degree of human devel-
opment of a population, defined as the range of choices offered in the course of life and the
freedom for individuals to enjoy the life they desire. This freedom and range of choices are
greatly influenced, of course, by the length of their lives, access to knowledge, and material
standard of living. The HDI is therefore a summary measure of three basic dimensions:
health, education, and income. The aggregate HDI, as well as its three dimensions, is repre-
sented by a number ranging from 0 to 1; the closer to 1, the greater the human development of
the population.

The Socioeconomic Index of the Geographic Context for Health Studies (GeoSES) aims to
evaluate and monitor health inequalities in Brazil. For this, it is based on the assumption that
the socioeconomic status (SES) is related to the prevalence of several diseases, the disease gra-
dient, and that the socio-economic context where individuals are inserted is essential to under-
stand health inequalities. In this sense, the index summarizes the living conditions in that
locality, separated into seven dimensions: education, poverty, wealth, income, racial segrega-
tion and deprivation, and services. The GeoSES Income index increases as the average wage of
the employed population increases. The GeoSES Education index decreases as the educational

attainment of the population increases. GeoSES Poverty decreases when the poverty of the population increases, there is an inversely proportional relationship. GeoSES Wealth seeks to represent the stock of resources in a given locality through real estate information, and the index grows as wealth increases in the municipality. The GeoSES Material Deprivation is measured through the material resources and conveniences available (such as adequate housing, ownership of a car, refrigerator, computer, etc.), also assessing their access to services such as sanitation, electricity, and internet. The higher the GeoSES Material Deprivation, the greater the material deprivation that a given municipality's population goes through. Finally, GeoSES Segregation considers aspects of inequality in access to education and income, including income and education levels stratified by ethnic groups. The higher the segregation in a municipality, the lower the GeoSES Segregation.

The fourth socioeconomic environment evaluated was made up of the variables Per Capita Household Income (average monthly family income per capita in 2010), Gini Index of household income, Illiteracy Rate (% of illiteracy in the population fifteen years old or older in 2010), Absence of Water and Sanitation (% of resident people without access to water and sewerage), Bolsa Família Program (BFP) Beneficiary (% of population benefited by the BFP), and Infant Mortality Rate (children under 1 year old per 1.000 live births), obtained from the 2010 census (IBGE/Brazil), and not directly considered in the composition of the composite indexes SVI, HDI, or GeoSES.

## Statistical modeling

We computed three distinct mortality rates (per 100,000 people) by municipalities in 2020: crude mortality rate (CMR), age and sex standardized mortality rate (SMR), and age and sex standardized mortality rate offset by the time of exposure to the epidemic (SMRT). CMR was calculated as the number of deaths from COVID-19 by the total population exposed to the risk (total municipality population) in 2010. SMR had computation similar to CMR, but each age-specific mortality rate by municipality was standardized (re-weighted) by using the Brazilian population pyramid (age and gender pyramid) in 2010 as reference. Lastly, SMRT is equal to SMR divided by the time of exposure to the epidemic, the last defined as the time (in months) since the first positive case of SARS-CoV-2 infection in each specific municipality.

To determine socioeconomic factors associated with mortality rates we used log-linear models adopting state codes as a random effect for the intercept. The three definitions of mortality rate were used as outcomes and the four groups of socioeconomic indexes as independent variables. Each of these groups were implemented separately in models to avoid highly correlation (see S1 Fig). The models were adjusted for Crowding Rate (%age of the population living in a house with more than two people per room in 2010), Demographic Density Rate (number of inhabitants per $km^2$ in 2010), State Capitals, and Travel Time to State Capital (in hours). We also did robustness checks using health infrastructure, proxied by ICU Hospital Beds Rate (the number of hospital beds in ICU per million inhabitants in 2015) and Physician Rate (number of physicians per thousand inhabitants in 2015) as controls. Measures of association were presented in terms of % change (exponentiated value of geometric mean) adjusted for confounders. The independence assumption among municipalities was loosened by including a Haversine variance-covariance matrix, where the dependence among municipalities (covariance) is computed by the spatial correlation. The Haversine formula calculated the geodesic distance between two municipalities using points specified by radian latitude/longitude.

We use the so-called "general and simple method" to calculate the two types of coefficients of determination (R2) of the adjusted LMMs, marginal and conditional R2, where the first

represents the proportion of variability explained by the fixed effects and the second represents the proportion of total variance explained through both fixed and random effects [22]. All statistical analyses were performed in R v.4.0.5 software, utilizing the 'nlme' and 'sf' libraries and their dependencies to develop statistical models and maps, respectively.

## Results

### Sample description

Of the 1,136,681 records present in the severe acute respiratory syndrome database (SIVEP-Gripe/MoH) as of January 11[th], 2021, 109 were eliminated for not having geographic information (country, state, city) of record origin, 2,681 for having a negative age, and 202 for being older than 105 years. In addition, in the remaining 1,133,689 records, the year, less than 2019 or greater than 2021, was corrected for dates of case notification, first symptoms, start and end of hospitalization, admission and discharge from intensive care units, clinical course, and death. Of the remaining 1,133,689 records, 607,864 (53.62%) had SARS-CoV-2 infection confirmed by laboratory, clinical, clinical-epidemiological, or clinical imaging (lung X-ray/computed tomography) criteria, and 191,528 (16.89%) had a death likely related to COVID-19 by that date. These deaths were distributed in 4,928 (88.55%) of the 5,565 Brazilian municipalities (IBGE); only 14 of these municipalities had recorded cases of COVID-19 without any deaths.

Fig 1 presents the distribution of the mortality rate among Brazilian municipalities according to the three different definitions, (A) CMR, (B) SMR, and (C) SMRT, as well as (D) a tabulation containing median values of these rates divided by Regions and sub-divided between state capital and non-capital municipalities, where maps stronger hues represent a higher mortality rate. This figure allows the observation of two different aspects of mortality in Brazil over the course of the 2020 COVID-19 pandemic, which must be considered together. The first is that, regardless of the definition of mortality rate, the figures are appropriate for assessing relative mortality among different regions and/or municipalities in the Brazilian territory. Second, is that the different mortality rate definitions allow us to evaluate changes in the increments of these rates given either by regional differences, or even, between state capitals and non-capital municipalities, of the age and gender distributions of the populations, or, no less discrepant, of the epidemic timing experienced by these municipalities/regions in the year 2020. Regardless of the death rate definition, the North region was the most affected, followed closely by the Midwest region, while the Northeast region was the least affected. The state capital municipalities were always more affected than non-capitals, either nationally or regionally. Apparently, the use of different death rate definitions seems to influence the relative severity among municipalities. For example, when comparing between the CMR (Fig 1a) and the SMR (Fig 1b) we notice a worsening in the latter in the North and Central-West regions, possibly due to the greater contribution of young people in the demographic composition of these regions. When we compare the SMR (Fig 1b) and the SMRT (Fig 1c), we see a decrease in severity, still high, in the coastal region, where most Brazilian state capitals are concentrated, possibly due to the longer duration of the epidemic in this region. The tabulation (Fig 1d) confirms these movements. There was an average 2.48-fold increase between CMR and SMR median mortality rates in the North and Midwest regions, while for the Northeast, Southeast and South regions this average increase was only 1.83-fold. Similarly, we observed an average increase of 2.82 times between the median mortality of capital and non-capital municipalities by the SMR mortality rate, while we observed an average increase of 1.86 times between the same municipalities using the median SMRT mortality rates.

Table 1 shows the median SMRT and interquartile ranges of the municipalities included in the top 10 and bottom 10 percentiles (10[th] or 1[st] decile) of different socioeconomic dimensions

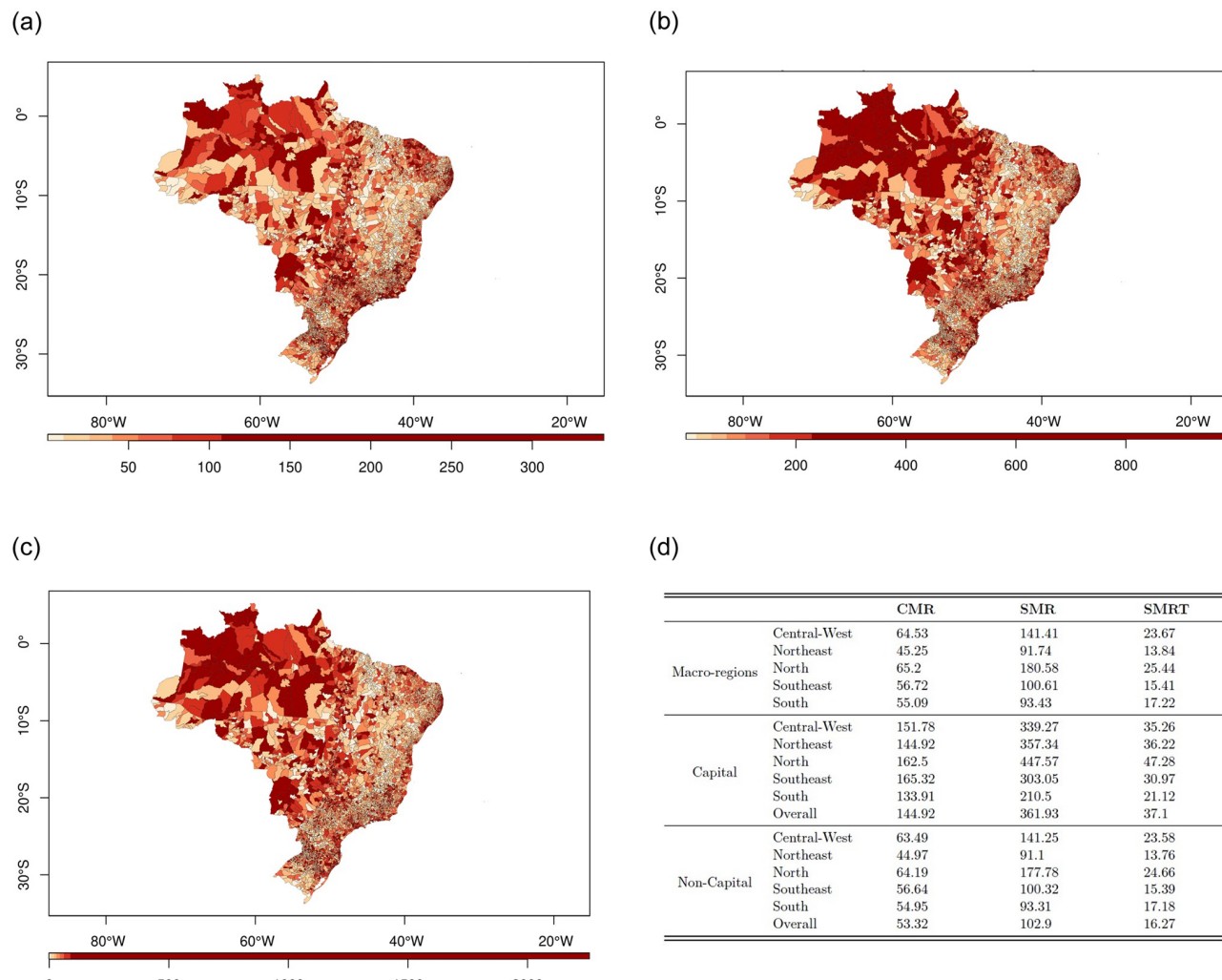

**Fig 1. Brazilian national map by municipalities with distribution of mortality rates (a, b and c) and medians of mortality rate by geographic locations (d).** Brazil, 2020. (a) CMR. (b) SMR. (c) SMRT. (d) Mortality Rate Medians.

of Brazilian municipalities. The Table focuses attention on the municipalities at the extremes of the distributions of socioeconomic dimensions, that is, how the top 10% municipalities differ from the bottom 10% considering each socioeconomic attribute in terms of SMRT. The inequality in the distribution of SMRT among Brazilian municipalities is evident when we look at the extreme deciles. The municipalities in the first SVI decile, or those with the highest social vulnerability, had a median SMRT of 18.46 (IQR = 17.26) while, at the other extreme, or the municipalities with the lowest social vulnerability, the median was 13.52 (IQR = 18.73). With respect to HDI, while the 10% worst municipalities in terms of human development had a median SMRT of 12.86 (IQR = 16.2), the 10% municipalities with the best human development indexes had a median SMRT of 19.95 (IQR = 16.47). Finally, the 10% municipalities with the lowest and highest socioeconomic status in the GeoSES had SMRT of 14.02 (IQR = 17.59) and 19.54 (IQR = 16.66), respectively. We observed in this preliminary examination, according to the three aggregate socioeconomic status indexes, that municipalities with higher socioeconomic levels had on average a higher SMRT than the relatively less developed municipalities, despite the large variability indicated by the wide IQR ranges. A similar situation was

**Table 1. Descriptive analysis of Brazilian municipalities by low (1ˢᵗ decile) and high (10ᵗʰ decile) SMRT, and socio-economic attributes.** Brazil, 2020.

| | Low SMRT median (IQR) | High SMRT median (IQR) |
|---|---|---|
| **Socioeconomic Variables** | | |
| SVI | 18.46 (17.26) | 13.52 (18.73) |
| SVI Urban Infrastructure | 18.72 (21.15) | 16.22 (20.23) |
| SVI Human Capital | 19.18 (16.12) | 14.69 (18.18) |
| SVI Labor and Income | 19.69 (16.53) | 11.75 (14.38) |
| HDI | 12.86 (16.2) | 19.95 (16.47) |
| HDI Education | 14.15 (16.44) | 20.1 (16.57) |
| HDI Health | 13.66 (14.77) | 18.24 (17.07) |
| HDI Income | 12.44 (15.3) | 19.51 (16.69) |
| GeoSES | 14.02 (17.59) | 19.54 (16.66) |
| GeoSES Education | 20.45 (16.03) | 13.67 (14.81) |
| GeoSES Poverty | 19.89 (16.52) | 13.52 (16.46) |
| GeoSES Deprivation | 14.25 (18.42) | 19.72 (17.72) |
| GeoSES Wealth | 16.88 (20.45) | 18.24 (16.51) |
| GeoSES Income | 11.64 (12.78) | 20.32 (17.7) |
| GeoSES Segregation | 14.48 (15.85) | 18.58 (18.58) |
| Household Per Capita Income | 13.09 (15.12) | 19.97 (17.27) |
| Gini Index | 19.81 (20.05) | 16.97 (23.53) |
| Illiterate Rate | 20.79 (16.99) | 12.08 (13.51) |
| Absence of Water and Sanitation | 17.64 (18.67) | 14.12 (19.11) |
| BF Recipients | 20.12 (16.75) | 12.74 (13.77) |
| Child Mortality Rate | 19.44 (20.48) | 16.48 (21.05) |
| **Control Variables** | | |
| Crowding Rate | 19.1 (19.95) | 18.63 (23.23) |
| Demographic Density Rate | 20.48 (25.54) | 24.42 (16.28) |
| Travel Time | 27.28 (19.61) | 13.73 (16.47) |
| Physician Rate | 18.84 (21.2) | 16.42 (16.21) |
| ICU Hospital Beds Rate | 15.82 (17.58) | 20.82 (16.47) |

monotonically observed for the socioeconomic status. For example, the first decile of munici-palities with the lowest inequality rates as measured by the Gini Index had a median of 19.81 (IQR = 20.05), while the tenth decile of municipalities with the highest inequality rates had a median of 16.97 (IQR = 23.53). It is worth noting that, as observed in the tenth deciles of the Gini, GeoSES Income and Household Per Capita Income Indexes, the largest interquartile ranges of the GeoSES Poverty, Deprivation and Segregation Indexes were also larger in the deciles representing better social conditions, first deciles for the latter, suggesting a greater var-iability in the SMRT distribution among the most developed Brazilian municipalities.

## Relationship between different socioeconomic dimensions and COVID-19-related mortality rate

Table 2 presents the estimated parameters in mixed-effects log-linear models and their mar-ginal and conditional coefficients of determination (R2) for the definitions of mortality rate (CMR, SMR, and SMRT), dependent variables, with the dimensions of the composite socio-economic indexes (SVI, HDI, and GeoSES) and other socioeconomic indicators, independent

Table 2. Adjusted log-linear models to estimate association between COVID-19 mortality rate (CMR, SMR, and SMRT) and socioeconomic status in brazilian municipalities. Brazil, 2020.

| | | | CMR | | | | | SMR | | | | SMRT | |
|---|---|---|---|---|---|---|---|---|---|---|---|---|---|
| | | (1) | (2) | (3) | (4) | (5) | (6) | (7) | (8) | (9) | (10) | (11) | (12) |
| SVI Human Capital | | -23.53** [-42.97; -4.09] | | | | -17.83 [-62.55;26.88] | | | | -0.07 [-0.43;0.30] | | | |
| SVI Labor and Income | | -62.87*** [-79.87; -45.86] | | | | -260.77*** [-299.68; -221.85] | | | | -1.10*** [-1.42; -0.79] | | | |
| SVI Urban Infrastructure | | -7.90 [-17.34;1.54] | | | | -5.54 [-27.10;16.03] | | | | -0.17* [-0.34;0.01] | | | |
| HDI Education | | | 28.29** [3.52;53.07] | | | | 79.86*** [23.33;136.39] | | | | 0.61** [0.14;1.07] | | |
| HDI Health | | | -12.97 [-62.46;36.53] | | | | -55.83 [-170.02;58.35] | | | | -0.89* [-1.83;0.04] | | |
| HDI Income | | | 176.74*** [139.21;214.28] | | | | 530.99*** [444.81;617.17] | | | | 1.95*** [1.24;2.65] | | |
| GeoSES Education | | | | -0.39*** [-0.67; -0.12] | | | | -1.10*** [-1.73; -0.47] | | | | -0.002 [-0.01;0.004] | |
| GeoSES Poverty | | | | -0.84*** [-1.04; -0.64] | | | | -2.34*** [-2.81; -1.88] | | | | -0.01*** [-0.02; -0.01] | |
| GeoSES Deprivation | | | | -0.30*** [-0.53; -0.08] | | | | -0.25 [-0.77;0.27] | | | | 0.002 [-0.003;0.01] | |
| GeoSES Wealth | | | | 0.31 [-1.79;2.41] | | | | -4.58* [-9.35;0.20] | | | | -0.11*** [-0.14; -0.07] | |
| GeoSES Income | | | | -0.001 [-0.01;0.004] | | | | 0.001 [-0.01;0.01] | | | | 0.0000 [-0.0001;0.0001] | |
| GeoSES Segregation | | | | -18.73 [-61.99;24.54] | | | | -130.31*** [-228.71; -31.92] | | | | -0.97** [-1.77; -0.16] | |
| Household Per Capita Income | | | | | 0.03*** [0.02;0.04] | | | | 0.09*** [0.06;0.11] | | | | 0.0002** [0.0000;0.0004] |
| Gini Index | | | | | -27.76*** [-48.24; -7.27] | | | | -83.15*** [-130.89; -35.41] | | | | -1.38*** [-1.77; -0.99] |
| Illiterate Rate | | | | | -0.27* [-0.55;0.02] | | | | -1.90*** [-2.56; -1.24] | | | | -0.02*** [-0.02; -0.01] |
| Absence of Water and Sanitation | | | | | -0.43*** [-0.58; -0.29] | | | | -1.13*** [-1.46; -0.79] | | | | -0.01*** [-0.01; -0.01] |
| BFP Recipients | | | | | -0.16** [-0.32; -0.002] | | | | -0.43** [-0.79; -0.07] | | | | 0.001 [-0.002;0.004] |
| Child Mortality Rate | | | | | -0.06 [-0.20;0.09] | | | | -0.06 [-0.39;0.27] | | | | -0.001 [-0.004;0.001] |

(Continued)

**Table 2.** (Continued)

| | | | | | | | | | | | | |
|---|---|---|---|---|---|---|---|---|---|---|---|---|
| | \multicolumn{12}{c}{*Dependent variable:*} | | | | | | | | | | | |
| | \multicolumn{4}{c}{CMR} | | | | \multicolumn{4}{c}{SMR} | | | | \multicolumn{4}{c}{SMRT} | | | |
| | (1) | (2) | (3) | (4) | (5) | (6) | (7) | (8) | (9) | (10) | (11) | (12) |
| Constant | 98.25*** | −60.30*** | 131.27*** | 75.99*** | 191.08*** | −273.52*** | 263.23*** | 136.17*** | 3.08*** | 1.69*** | 3.26*** | 3.44*** |
| | [88.34;108.15] | [−96.54; −24.07] | [107.82;154.72] | [61.02;90.96] | [162.52;219.65] | [−358.62; −188.42] | [207.49;318.97] | [97.57;174.77] | [2.89;3.27] | [1.01;2.38] | [2.82;3.70] | [3.16;3.72] |
| Marginal $R^2$ [2] | 0.120 | 0.154 | 0.128 | 0.169 | 0.087 | 0.103 | 0.085 | 0.134 | 0.043 | 0.042 | 0.053 | 0.099 |
| Conditional $R^2$ | 0.291 | 0.336 | 0.289 | 0.366 | 0.484 | 0.519 | 0.456 | 0.544 | 0.228 | 0.230 | 0.230 | 0.310 |

Notes:

[1]

*$p<0.1$;

**$p<0.05$;

***$p<0.01$.

[2] Based only on socioeconomic variables.

[3] All adjusted models included Crowding Rate, Demographic Density Rate, State Capitals and Travel Time to State Capital as confounders.

variables; totaling twelve distinct models (Table 2 (1)-(12)), of same random effect and same spatial dependence structure.

In order to deepen the relationship between the various socioeconomic dimensions/indicators of the Brazilian municipalities with the mortality rate, we chose both to use the disaggregated version of the composite indexes, and to focus our presentation of results on the SMRT mortality rate, as we consider it more "fair"; since it corrects and standardizes the age distribution of the population by sex, as well as the pandemic time expected by each municipality. The results with the CMR and SMR definitions are used in comparison to the SMRT when relevant.

The socioeconomic dimension of education (Table 2), HDI Education, we observed a direct and significant (P-value < 0.05) relationship with all the mortality rate definitions, i.e., the more educated the municipality's population was, the higher its COVID-19 mortality rate was; albeit independently of the age distribution by sex or the pandemic time expected by that municipality. Consistently, the lower a municipality's Illiteracy Rate was the higher the observed SMRT was (P-value < 0.01). For two other composite index dimensions intrinsically associated with education, SVI Human Capital and GeoSES Education, we observed signs of the mean negative and positive adjusted coefficients, respectively, consistent with those previously cited, however, not significant in models with the SMRT outcome. Remember that, while an increase in the value of any SVI dimension indicates greater vulnerability, the higher the observed value of a municipality's GeoSES Education index, the lower the expected level of schooling for its population.

The socioeconomic dimension of income and poverty (Table 2), in general, we observed that the lower-income municipalities were not those for which we observed the highest death rate in the year 2020, even though controlled by the other socioeconomic dimensions of the same composite indices and by factors intrinsic to the Sars-Cov-2 spread, that is, the distance between the observed municipality and the state capital (common origin of the pandemic in each state) or between nearby municipalities, population density, political and climatic conditions, and other characteristics of each Brazilian state. The increase in SVI Income and Labor was indirect and significant (P-value < 0.01) related to the increase in SMRT. Consistently, increases in Household Per Capita Income and HDI Income were directly related (P-value < 0.01) to increases in SMRT. Recall that these three results indicate that the higher the income, i.e., the lower the SVI Income and Labor, and the higher the Household Per Capita Income and HDI Income, the higher the SMRT corrected death rate. The composite GeoSES index includes the dimensions GeoSES Income, directly associated with income, and two other GeoSES Wealth and Poverty more associated with poverty. While for the former we observed no significant relationship with SMRT, increasing GeoSES Wealth was indirectly related to increasing SMRT (P-value < 0.01). The same, indirect relationship, was observed for GeoSES Poverty not only with significantly increased SMRT but also with that of CMR and SMR (P-values < 0.01). These two dimensions of GeoSES agree that the higher the wealth or lower the poverty of a municipality, the lower the average COVID-19-related mortality rate experienced by that municipality in Brazil in 2020. Finally, we observed no significant relationship between the BFP Recipient population, published by IBGE, and the SMRT.

Still related to income and poverty, but measuring intraregional or inter-municipal inequality of these two, we respectively observed the relationships of both the Gini index, published by IBGE, and the Segregation dimension of the GeoSES composite index with the different definitions of mortality rate (Table 2). The increase in both was related indirect and significantly with the increase in SMRT (P-value < 0.01). That is, not only did richer and less poor municipalities have higher SMRT on average, but those with lower income inequality and/or poverty were the ones that experienced higher COVID-19-related death rates. Recalling that

the higher, or closer to 1, is the Gini index the greater the inequality in income distribution, and similarly, the higher the GeoSES Segregation measure the less segregation, understood as the separate housing of different population groups in different parts of a city or segregation areas, observed in that municipality.

Regarding the dimensions directly associated with health and infrastructure (Table 2), the HDI Health index and SVI Infrastructure, we observed conflicting and significant results only after corrections and standardization by the population age distribution by sex and by the pandemic time expected by each municipality, i.e., for the SMRT outcome and not for the CMR and SMR outcomes. We observed both an indirect relationship with the SMRT. That is, the better the health status, or even, the higher the life expectancy experienced in a municipality, the lower on average was the observed SMRT (P-value < 0.01), whereas the higher the vulnerability of a municipality's infrastructure, that is, the lower the conditions of access to basic sanitation and urban mobility services, the lower, on average, was the observed SMRT. We did not observe a significant relationship between SMRT and GeoSES Deprivation, which very close to SVI Infrastructure would indicate the deprivation of material resources and essential services of a municipality, thus we cannot assess the robustness of this observation; we can only state that the higher the GeoSES Deprivation the lower was its crude mortality rate, or CMR (P-value < 0.01), to some degree corroborating the SVI Infrastructure results. Remembering that for any dimension of SVI, the greater the increase in the value of GeoSES Deprivation, the greater the deprivation of the measured region.

This much said, when we analyze the values of the coefficients of determination, $R^2$, especially those related to the socioeconomic dimensions/indicators of the Brazilian municipalities, that is, the marginal coefficient of determination of each log-linear mixed effect model, we have results that vary from 0.042 (4.2%), obtained with the inclusion of the socioeconomic dimensions of the HDI composite indicator, up to 0.169 (16.9%), obtained with the inclusion of the variables Per Capita Household Income, Gini Index of household income, Illiteracy Rate, Absence of Water and Sanitation, Bolsa Família Program (BFP) Beneficiary, and Child Mortality Rate, obtained from the 2010 census (IBGE/Brazil) (Table 2).

While we showed robust correlations between the socioeconomic characteristics of Brazilian municipalities and the COVID-19-related mortality rate in 2020, we also found that the average of the socioeconomic data of Brazil's municipalities explains very little in the COVID-19-related mortality rate in Brazil.

## Discussion

We evaluated the importance of municipal socioeconomic factors as a determinant of COVID-19-related mortality rate in Brazil in 2020. More than that, we compared different definitions of mortality rate, where the first was the crude mortality rate, the second was the sex- and age-standardized mortality rate and the third was the sex- and age-standardized mortality rate weighted by the time of the pandemic. To our knowledge, this is the first study to compute and compare diverse COVID-19-related mortality rates at the municipal level in Brazil. In fact, we have not found any other study, in Brazil or in any other country, that has compared results obtained from different definitions, adjusted or not, of mortality rates in regional groupings. The use and comparison of these distinct definitions of mortality rate brought important evidence. When we excluded the effect of demographic structure and pandemic timing from the calculation of mortality rates, the contribution of socioeconomic factors to explain differences in COVID-19-related mortality rates became insignificant. The correlation of the elderly population (a population at very high risk of COVID-19-related death) to the total population and socioeconomic factors were partially responsible for the $R^2$ found in the crude mortality rate

models, and probably not their intrinsic ability to change the population's exposure to infection. Furthermore, the progressive spread of the SARS-CoV-2 virus happened at different speeds and municipalities were at different stages during the pandemic. Hence, weighting by the pandemic time experienced by each municipality partially explains the $R^2$, since usually the state capitals were the first to be affected by the COVID-19 pandemic in Brazil. Roubaud *et al.* (2020), the only study that used the entire national data and also included socioeconomic factors under consideration, confirmed our results considering the $R^2$ coefficient [18].

Our first finding, and the answer to the question indicated in the title, was that socioeconomic factors explained very little of the differences in the municipality-level mortality rate observed in the COVID-19 pandemic in Brazil in 2020. Despite the low ability of socioeconomic factors to explain the mortality rate, we used the model parameters to analyze the influence of each dimension of the socioeconomic status on the COVID-19-related mortality rate in Brazilian municipalities in 2020. We obtained strong evidence pointing out that municipalities with higher income (SVI Labor and Income, HDI Income and Household Per Capita Income), better education (HDI Education and Illiterate Rate), and urban infrastructure (SVI Urban Infrastructure and Absence of Water and Sanitation) had higher COVID-19-related mortality rates on average when compared to municipalities on average poorer and lacking these resources. It is crucial to highlight that this result is independent of Crowding or Density Rate, healthcare coverage (Physician Rate and ICU Hospital Beds Rate), distance to state capital (Travel Time), urban infrastructure, poverty, income inequality or the distance among municipalities (Haversine Var-Cov matrix), and is repeated among close SES approaches (SVI, HDI and GeoSES). The relationship between the BFP Recipient population and the SMRT may somehow indicate the efficiency of social security programs, that is, we could not interpret it as increased in poorer municipalities, but rather interpret it as municipalities where the social safety net is more efficient regardless of higher or lower income.

Similar result for income, but not for education, was found by Roubaud *et al.* (2020) also in a study grouped by Brazilian municipalities [18]. At this point it is interesting to remember that SARS-CoV-2 infections first occurred in capital cities, characterized by higher per capita income, more unequal income distribution, and more pronounced social disparities compared to non-capital municipalities. The virus has progressively interiorized, possibly via the dense urban network that connects capital cities to other municipalities. Consequently, the lack of urban infrastructure, which also includes mobility and transportation structure, influenced the delayed spread of SARS-CoV-2 to small, towns with less urban infrastructure and also lower per capita income. Consistent with this narrative, we also found evidence that more unequal municipalities had fewer COVID-19-related deaths on average.

Undoubtedly, social and economic factors, as already pointed out in several studies, had a relationship with COVID-19-related deaths in Brazil in 2020 [12–17]. Lower per capita income and education, black/brown skin color, household conditions leading to crowding, informal occupation, and specific economic activities were found to be risk factors for COVID-19-related infection and death. However, it is important to note that these studies were based on intra-municipal data, i.e., either individual or grouped by neighborhoods. Our results show that it is not possible to extrapolate these results to municipal aggregates, nor the other way around. Certainly, extrapolation of our results, aggregated by municipalities, to individuals would have a large ecological bias or imply an ecological fallacy. High-income municipalities generally had higher mortality rates due to COVID-19, but it is not possible to say that the wealthier individuals were the most affected and died the most. There is the possibility that the poorer population of the wealthier municipalities, perhaps less able to adhere to social distancing policies, and therefore at higher presumed risk to infection, died the most in these municipalities. This seems to be the case when we analyze the

association between poverty (GeoSES Poverty) and racial segregation (GeoSES Segregation) with mortality rate. Our results show that the higher the deprived population and the more racially polarized the municipality was, the higher the mortality rate was also. Racial inequality and poverty were significant even when considered together with education and income, and their effects remained even when health infrastructure was included in the model. Roubaud *et al.* (2020) also found that racial minorities, such as African Americans, were more affected even when controlling for income effects on the indicators [18]. In fact, this result was not only reported for Brazil [23, 24]. Roubaud *et al.* (2020) also found that the greater presence of communities concentrating deprived populations and African Americans, *favelas* or subnormal clusters, was associated with higher SMRT [18]. Although not our primary goal, we can say that the effect of racial inequality on COVID-19-related mortality rate across municipalities was not driven by income, urban condition, education, or health coverage because we used these as controls in the models. The causal mechanism may be some factor that was not considered in our statistical models, and further work would be needed to elucidate this. On the other hand, as for the relationship between pre-existing health condition, embodied in HDI Health, and mortality rate, the results suggested that favorable pre-existing health condition can reduce the probability of death in the municipalities, and that the other way around, poor pre-existing health condition, would be therefore a risk factor to increase COVID-19-related death susceptibility (Table 2), column 10).

The level of analysis is a classic issue in epidemiology. Aggregation into municipalities, an ecological approach, poses the dilemma of using equal or proportional weights to demographics when evaluating associations in models. This problem is not solved solely by using mortality rates, although this implies an important demographic correction. In this study we chose to consider equal weights for the different municipalities because, from a legal point of view, regardless of their population or geographical size, each unit has the same status vis-à-vis the law and their rights. Among COVID-19 studies, we found evidence that the impact of SES on COVID-19-related mortality may vary across levels of aggregation. Knittel and Ozaltun (2020) found, for example, differences across states and counties in the significance of race and mobility [25].

It is also interesting to note that the narrative of the pandemic at the municipal level in Brazil differs from others observed in high-income countries, where, the impact of the pandemic on mortality was greater in poorer municipalities [10]. The particular social condition of Brazil, the very unequal distribution of wealth and opportunities among different regions, and the concentration of economic activity circuits in the capital cities, may partly explain these differences.

Since our results revealed that socioeconomic determinants did little to explain the increase in COVID-19-related death rates in Brazil in 2020, the question remains as to what these determinants might be. We speculate that a possible determinant would be the lack of federal coordination of public policies agreed with states and municipalities, such as the delimitation of sanitary barriers to reduce the inter-municipal movement of people by public and private transport, or even greater sanitary control of the network that connects municipalities by means of transport, services, and commerce. Another possible factor would be the level of economic development of the municipalities and, consequently, their economic structures. These structures can condition the sector of activity composition and the employment relationship (formal/informal jobs), influencing the worker's ability to adhere to social distancing policies and reduce their exposure to SARS-CoV-2. Thus, a structural economic precondition may exist, which would pose inequalities among municipalities and influence the epidemic burden and mortality.

The robustness of the estimates was a constant concern in this study. To deal with this issue, we took several precautions. First, the mortality rate was standardized by sex and age and weighted by the time of the pandemic. Second, we used the Haversine variance-covariance matrix in all regressions to capture the similarities of nearby municipalities in the significance of the parameters. Third, all models used Crowding Rate and Demographic Density Rate to deal with the natural causes of virus spread due to urban organization, and in addition, Travel Time to State Capital to control for the ease of access of its residents to major (capital) cities. Fourth, additional models were fitted for Hospital Bed Rate, ICU Hospital Bed Rate, Physician Rate, and Nurse Practitioner Rate to control for the possibility of differences in death among counties due to limited health service coverage (S1 Table). These additional controls for health service coverage did not change the significance of the $R^2$ coefficient of the SES and were dropped from further analysis. Sixth, we attempted to work with socioeconomic variables categorized by deciles (first, tenth, and second to ninth) to observe some difference in the coefficients when compared to the numerical possibility. Again, no substantial discrepancies were found, showing the robustness of the results (S2 Table). We found great robustness in our results. We used different composite indicators aiming to represent similar socioeconomic dimensions. In this way, different specifications were used to verify the sensitivity of the socioeconomic dimensions to changes in the composition of the indicators. In our results, both the significance and the sign related to the analyzed socioeconomic dimensions change little when we change the indicators selected to represent them.

There are significant limitations to this study. These can be separated into: (i) Lack of available information for some questions in national/municipal perspective; (ii) Specificities of the COVID-19 epidemic data used; and, (iii) Lack of approach to infer causality. Concerning (i), we did not include in the models, meteorological conditions such as temperature, humidity, and rainfall, which have been mentioned in previous studies as a natural cause to influence the spread of SARS-CoV-2 infection [20, 26–28]. There is a lack of information available on this subject at the national/municipal level in Brazil. However, we included the state of belonging of each municipality as a random effect in the models to control for, among others, these differences in regional weather conditions. Another limitation is the availability of only outdated socioeconomic information, belonging to the last Brazilian census, in 2010. All socioeconomic index groups, HDI, SVI, GeoSES, and those selected from the census itself used the same primary information from the 2010 census in their definitions. It may introduce bias to SES and mortality rate distribution among municipalities, although there was no significant change in SES status, internal migration or population growth to alter their relative distribution among the almost 5,000 municipalities. Concerning (ii) only hospital deaths were considered and with this COVID-19 related death information is absent for numerous records, not to mention that because of low testing for SARS-CoV-2 infection throughout 2020, especially during the early stages of the pandemic, we may have underestimated or even distorted COVID-19 related mortality rates. We addressed these limitations by including patients with a confirmatory diagnosis of SARS-CoV-2 infection and who had death known to be related to COVID-19. In addition, we use controls in the model estimates to account for differences in health care resources across counties. Due to the characteristics of the data available, we carried out an ecological study. In several situations, the use of the data at an individual level can provide more reliable information for the development of public health policies. Nevertheless, as the differences between the ecological/individual analysis were extensively addressed in this study, it should be considered more of a point for debate than a limitation. Concerning (iii), we chose a cross-sectional study that did not allow us to analyze causal-type relationships between COVID-19-related mortality rate and socioeconomic factors. Previous studies have used the natural experiment design to analyze causality, but only using strong assumptions that are not

valid in Brazil, such as the uniformity of social distancing policies across the national territory [10].

## Conclusion

Our research concluded that, consistent across all socioeconomic indexes tested, high-income, more educated, and well infrastructure municipalities generally had higher mortality rates, although they did little to explain the increase in COVID-19-related mortality rates in Brazil in 2020. We recognize, however, that these results cannot be directly reflected in the reality of individual and/or intra-municipal clusters. In the future, contingency programs should focus on coordinating policies for social isolation and inter-municipal mobility, such as the delimitation of sanitary barriers to reduce the inter-municipal movement of people, or a sanitary control of the network that connects municipalities. More studies are needed to explore other possible determinants of differences in COVID-19-related mortality rates across municipalities, especially social isolation policies.

## Supporting information

**S1 Fig. Correlation matrix of independent variables in the models.** Dark and light, red and blue circles represent the intensity of correlations.
(TIF)

**S1 Table. Robustness check of adjusted log-linear models to estimate association between COVID-19 mortality rate (CMR, SMR and SMRT) and socioeconomic status in Brazilian municipalities.** Brazil, 2020.
(PDF)

**S2 Table. Description of the dimensions of the three socioeconomic status approaches, SVI, HDI, and GeoSES.**
(PDF)

**S3 Table. Patients description.**
(PDF)

## Author Contributions

**Conceptualization:** Julio Castro-Alves, Marcelo Ribeiro-Alves.

**Data curation:** Julio Castro-Alves, Lídia Santos Silva, João Paulo Lima, Marcelo Ribeiro-Alves.

**Formal analysis:** Julio Castro-Alves, Lídia Santos Silva, João Paulo Lima, Marcelo Ribeiro-Alves.

**Funding acquisition:** Julio Castro-Alves, Marcelo Ribeiro-Alves.

**Investigation:** Julio Castro-Alves, Lídia Santos Silva, João Paulo Lima, Marcelo Ribeiro-Alves.

**Methodology:** Julio Castro-Alves, João Paulo Lima, Marcelo Ribeiro-Alves.

**Project administration:** Julio Castro-Alves, Marcelo Ribeiro-Alves.

**Resources:** Julio Castro-Alves, Marcelo Ribeiro-Alves.

**Software:** Julio Castro-Alves, Marcelo Ribeiro-Alves.

**Supervision:** Julio Castro-Alves, Marcelo Ribeiro-Alves.

**Validation:** Julio Castro-Alves, Lídia Santos Silva, Marcelo Ribeiro-Alves.

**Visualization:** Julio Castro-Alves, Lídia Santos Silva, Marcelo Ribeiro-Alves.

**Writing – original draft:** Julio Castro-Alves, Lídia Santos Silva, João Paulo Lima, Marcelo Ribeiro-Alves.

**Writing – review & editing:** Julio Castro-Alves, Lídia Santos Silva, Marcelo Ribeiro-Alves.

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
