## [Decision Letter · Decision Letter 0]

4 Jan 2022

PONE-D-21-24701Were the socio-economic determinants of municipalities relevant to the increment of COVID-19 related deaths in Brazil in 2020?PLOS ONE

Dear Dr. Castro-Alves,

Thank you for submitting your manuscript to PLOS ONE. After careful consideration, we feel that it has merit but does not fully meet PLOS ONE’s publication criteria as it currently stands. Therefore, we invite you to submit a revised version of the manuscript that addresses the points raised during the review process.

Specifically, please address all the issues raised in the attached pdf file as well as all those raised by both reviewers. The manuscript has a lot of grammatical errors some of which have been indicated in the attached pdf file. Please carefully correct all grammatical errors remembering that not all the errors have been pointed out in the review comments. It is the authors' responsibility to ensure that the manuscript is well written and in correct English. Please consider having the manuscript reviewed by a native English speaker.

We look forward to receiving your revised manuscript.

Kind regards,

Agricola Odoi, BVM, MSc, PhD, FAHA, FACE, Dipl. AVES (Hon)

Academic Editor

PLOS ONE

Journal Requirements:

“This research was funded by Inova-Fiocruz Program: “Programa Fiocruz de Fomento `a Inova¸c˜ao - Gera¸c˜ao De Conhecimento - Enfrentamento da Pandemia e P´os-pandemia COVID-19 Encomendas Estrat´egicas” (Grants: VPPCB-005-FIO-20-2-45 and VPPCB-005-FIO-20-2-61).”

 “This research was funded by Inova-Fiocruz Program (VPPCB-005-FIO-20-2-45, https://portal.fiocruz.br/programa-inova-fiocruz) to M. Ribeiro-Alves and Inova-Fiocruz Program (VPPCB-005-FIO-20-2-61, https://portal.fiocruz.br/programa-inova-fiocruz) to J. Castro-Alves. The funders had no role in the conceptualization, analysis, interpretation, or decision to publish this manuscript.”

Reviewers' comments:

Reviewer's Responses to Questions

**Comments to the Author**

1. Is the manuscript technically sound, and do the data support the conclusions?

Reviewer #1: Partly

Reviewer #2: Yes

2. Has the statistical analysis been performed appropriately and rigorously? 

Reviewer #1: Yes

Reviewer #2: Yes

3. Have the authors made all data underlying the findings in their manuscript fully available?

Reviewer #1: Yes

Reviewer #2: Yes

4. Is the manuscript presented in an intelligible fashion and written in standard English?

Reviewer #1: No

Reviewer #2: Yes

5. Review Comments to the Author

Reviewer #1: The authors investigated whether socioeconomic determinants of municipalities are associated with

COVID-19 death rates in Brazil in 2020. Below are my comments:

General comments

1. Area-based indicators of SES are not as robust as the individual-based indicators of SES. While the use of different definitions of mortality rate and SES indicators is useful, given that there are already several studies on this topic using the more accurate individual/person-based indicators of SES, the authors need to clearly discuss why this ecological study is important and the implication of the findings. What novel information does the study add to the existing literature. Why is understanding the relationship between SES and the COVID-19 mortality rate from an ecological perspective important?

2. A major limitation of the study that has not been discussed (but was only mentioned) is the use of outdated indicators of SES to assess association with the mortality rate of a disease that occurred almost 10-years later. Do these SES indicators represent the current circumstances? If these SES indicators have not changed in the last 10 years, this should be mentioned or at least discuss possible implications of using outdated indicators on the current findings.

3. Another limitation that should be included is that use of the 2010 population census to calculate mortality rate would possibly overestimate the rates as the population would have increased between 2010 and 2020.

4. The manuscript requires proof reading and editing to correct typographical and grammatical errors to improve the flow and readability, e.g., please use ‘infection’ (or something relevant) instead of ‘contamination’.

Specific comments

Methods

1. Lines 156-158, are there patients whose COVID-19 diagnosis was determined by imaging only or was this in combination with laboratory tests?

2. Line 162, registry should be registrant.

3. Lines 253-255, please revise the sentence to include the method used

Results

1. Please provide overall descriptive data (number, percentage) on patient/registrant characteristics e.g., mean age, age group, sex, region, capital/non-capital and other available relevant characteristics.

2. Please write the results clearly and succinctly for the reader to understand without going back to read up on the different SES indicators and avoid unnecessary repetition, e.g., it is not necessary to use both ‘direct and positive coefficients’ or ‘inverse and negative coefficients’ for the direction of the association between SMRT and SES.

Also, instead of adding the sentences that begin with remember/recall…..you could just write e.g., for lines 371-372: ‘The increase in SVI Income and Labor, which indicates socioeconomic disadvantage, was inversely (P-value < 0.01) associated with increase in SMRT’. Please make the necessary revisions in the results section.

3. Line 274, please include the number of the figure referred to

4. Figure 1, please include a brief description of the colour gradients for the legend in terms of low/high mortality rate

5. Table 1, for ease of understanding, can the two columns (1st and 10th) be labelled as low vs high SES or most disadvantaged vs most advantaged?

6. Lines 344-350, this text can be moved to the Methods section under statistical analysis.

7. Line 354-355, note that the outcome was age and sex adjusted with adjustment for the pandemic time, however, these variables were not included in the models as independent variables.

8. Please check the interpretation of the findings on lines 383-386, it does not seem to align with the definitions of the GeoSES Poverty and GeoSES Wealth provided in lines 208-211.

9. Move the sentence on lines 387-390 to the Discussion section

10. Lines 391-402, if increasing Gini index, represents disadvantage, and is inversely associated with SMRT, it means that the disadvantaged experienced lower COVID-19-related death rates. Is this right?

11. Lines 429-433, please revise this sentence so the message is clear.

Discussion

1. Lines 500-504, these findings contradict the main study findings of higher SMRT among socioeconomically advantaged communities. The contradictions in these findings should be discussed so the reader can understand why this is the case.

2. Lines 514-519, is this speculation or is there evidence for this discussion?

3. Lines 558-561, this is the first mention of the sensitivity analyses, please include a sentence about these in the Methods and Results sections.

4. Lines 563-566, please provide the results for these analyses as a supplementary for better understanding of the statements.

5. Line 567, delete ‘Our concern apparently paid off’.

6. Please include a brief discussion about limitations or caveats associated with ecological studies and how these might affect the current findings.

Conclusion

1. Lines 603-605, ‘………………………., they did influence the promotion of contamination’. Please note that the study does not provide data to support this statement. The conclusion should include information that is supported by the study.

Reviewer #2: I have enjoyed reading your article. Please see my comments below.

Please clarify and discuss the following statement:

“At first, elites and high socioeconomic status (SES) communities would become infected because they were not yet prepared to deal with the emerging disease” Could you discuss and describes why this would be the case? Why high SES groups would have higher risk at the beginning of the epidemic? Perhaps they are more mobile, travel more, have higher interaction with other people, etc.

Please discuss more why at the later stages of the COVID -19 epidemic low SES groups had higher infection rates. Perhaps lower access to vaccines, differences in jobs (they must work and cannot stay home to prevent infections, household structure (e.g. more than one generation leaves in a house or more crowded households)

Why did you use a log-linear model? You could consider Poisson or Negative binomial model where the outcome variable would be the number of deceased in a municipality and accounting for the background population as an offset. Alternatively, you could build a Cox Proportional-Hazards Model to investigate the association between the survival time (or the time between exposure and death) of COVID19 patients and SES predictor variables.

Thank you for considering my comments.

6. PLOS authors have the option to publish the peer review history of their article (what does this mean?). If published, this will include your full peer review and any attached files.

Reviewer #1: No

Reviewer #2: No

---

## [Author Response · Author response to Decision Letter 0]

24 Feb 2022

February 10th, 2022

Dear Dr. Agricola Odoi

Academic Editor

PLOS ONE 

We would like to thank all the reviewers for the manuscript’s helpful comments and suggestions. We thoroughly revised the manuscript and addressed the reviewer’s comments. In addition, we have also taken the recommendations made by the editor in the form of comments in the original document. Changes in the manuscript are indicated in this letter, and in an uploaded new manuscript’s version with changes tracked. A point-by-point response to the reviewers’ comments is provided below. Comments are in regular black font, the author’s answers in standard red font, excerpts from the editions (mostly inclusions) in the manuscript, in italics.

Please feel free to contact us with any questions or concerns. We look forward to hearing from you.

Sincerely,

Julio Castro-Alves, PhD

Fundação Oswaldo Cruz, FIOCRUZ

Av. Brasil, 4365 - Manguinhos, ZIP: 21040-360, Rio de Janeiro, RJ, Brazil.

Phone: +55 21 99907-9192

E-mail: julio.castro.alves.lima@gmail.com

---

## [Decision Letter · Decision Letter 1]

15 Mar 2022

Were the socio-economic determinants of municipalities relevant to the increment of COVID-19 related deaths in Brazil in 2020?

PONE-D-21-24701R1

Dear Dr. Castro-Alves,

We’re pleased to inform you that your manuscript has been judged scientifically suitable for publication and will be formally accepted for publication once it meets all outstanding technical requirements.

Kind regards,

Agricola Odoi, BVM, MSc, PhD, FAHA, FACE, Dipl. AVES (Hon)

Academic Editor

PLOS ONE

Additional Editor Comments (optional):

Reviewers' comments:

Reviewer's Responses to Questions

**Comments to the Author**

1. If the authors have adequately addressed your comments raised in a previous round of review and you feel that this manuscript is now acceptable for publication, you may indicate that here to bypass the “Comments to the Author” section, enter your conflict of interest statement in the “Confidential to Editor” section, and submit your "Accept" recommendation.

Reviewer #1: (No Response)

Reviewer #2: All comments have been addressed

2. Is the manuscript technically sound, and do the data support the conclusions?

Reviewer #1: Yes

Reviewer #2: Yes

3. Has the statistical analysis been performed appropriately and rigorously? 

Reviewer #1: Yes

Reviewer #2: Yes

4. Have the authors made all data underlying the findings in their manuscript fully available?

Reviewer #1: Yes

Reviewer #2: Yes

5. Is the manuscript presented in an intelligible fashion and written in standard English?

Reviewer #1: Yes

Reviewer #2: Yes

6. Review Comments to the Author

Reviewer #1: (No Response)

Reviewer #2: (No Response)

7. PLOS authors have the option to publish the peer review history of their article (what does this mean?). If published, this will include your full peer review and any attached files.

Reviewer #1: No

Reviewer #2: No

---

## [Editor Report · Acceptance letter]

20 Apr 2022

PONE-D-21-24701R1 

Were the socio-economic determinants of municipalities relevant to the increment of COVID-19 related deaths in Brazil in 2020? 

Dear Dr. Castro-Alves:

I'm pleased to inform you that your manuscript has been deemed suitable for publication in PLOS ONE. Congratulations! Your manuscript is now with our production department. 

Kind regards, 

on behalf of

Prof. Agricola Odoi 

Academic Editor

PLOS ONE